# ST$_k$: A Scalable Module for Solving Top-k Problems[*]

**Hanchen Xia**[†,♣], **Weidong Liu**[†,‡], **Xiaojun Mao**[†,§]
{[†]School of Mathematical Sciences,
[‡]Ministry of Education Key Lab of Artificial Intelligence,
[§]Ministry of Education Key Laboratory of Scientific and Engineering Computing}
Shanghai Jiao Tong University, Shanghai, China
♣RoyalFlush AI Research Institute, Hangzhou, China
{x_hc_2000, weidongl, maoxj}@sjtu.edu.cn

## Abstract

The cost of ranking becomes significant in the new stage of deep learning. We propose ST$_k$, a fully differentiable module with a single trainable parameter, designed to solve the Top-k problem without requiring additional time or GPU memory. Due to its fully differentiable nature, ST$_k$ can be embedded end-to-end into neural networks and optimize the Top-k problems within a unified computational graph. We apply ST$_k$ to the Average Top-k Loss (AT$_k$), which inherently faces a Top-k problem. The proposed ST$_k$ Loss outperforms AT$_k$ Loss and achieves the best average performance on multiple benchmarks, with the lowest standard deviation. With the assistance of ST$_k$ Loss, we surpass the state-of-the-art (SOTA) on both CIFAR-100-LT and Places-LT leaderboards.

## 1 Introduction

The ranking problem is quite common in the field of AI. For imbalanced datasets, the Average Top-k (AT$_k$) Loss is more suitable than the conventional Average Loss [Lyu et al., 2020]. In the context of ambiguous classification tasks, Top-k Learning allows the ground truth to fall within the largest $k$ probabilities, enhancing the model's generalizability [Lapin et al., 2017, Berrada et al., 2018, Petersen et al., 2022]. For language models, the Top-k sampling method helps the models select the top $k$ most probable words during text generation, producing more fluent and coherent sentences. Distributed learning systems employ Top-k sparsification with error compensation (Top-k SGD) to reduce communication traffic without noticeably impacting model accuracy [Chen et al., 2018, Lin et al., 2018, Shi et al., 2019]. However, as deep learning models continue to grow in size, **the cost of the ranking process becomes increasingly significant**. For example, on a single NVIDIA H800 GPU, tuning a Llama-8B model using the LoRA [Hu et al., 2022] method with a batch size of 4096 and 1000 iterations takes 990.14 seconds. In particular, performing QuickSort on the individual losses consumes 86.33 seconds.

In this work, we propose ST$_k$ (Smoothed Top-k), a scalable module for Top-k problems. By adding only a single trainable parameter, ST$_k$ is able to solve the Top-k problem in $\mathcal{O}(n + k)$ steps. Due to the fully differentiable nature of ST$_k$, it can be embedded end-to-end as a layer. Experiments show that we can even add $\lambda$ to the computational graph for unified optimization using Stochastic Gradient Descent (SGD), which means that we do not need to consider the time cost of solving the Top-k problems within the computational graph. However, we can still enjoy the performance improvements that Top-k optimization provides. The contributions of this work are summarized as follows:

---

[*]Weidong Liu and Xiaojun Mao are the co-corresponding authors. Xiaojun Mao's research is supported by NSFC Grant No. 12422111 and 12371273, the Shanghai Rising-Star Program 23QA1404600 and Young Elite Scientists Sponsorship Program by CAST (2023QNRC001).

38th Conference on Neural Information Processing Systems (NeurIPS 2024).

- We propose a uniformly convergent approximation of the ReLU function.
- We propose an efficient and robust Smoothed Top-k module, $\mathrm{ST}_k$.
- We apply $\mathrm{ST}_k$ Module to smooth $\mathrm{AT}_k$ Loss, resulting in performance improvement and refreshing the state-of-the-art methods on two long-tailed learning leaderboards.
- We design an imbalanced classification dataset with a theoretical decision boundary.

For experiments, by applying the $\mathrm{ST}_k$ Module, we smooth the $\mathrm{AT}_k$ Loss into $\mathrm{ST}_k$ Loss. The computation time of $\mathrm{ST}_k$ Loss is almost identical to that of the average aggregating method, significantly faster than the sorting-required $\mathrm{AT}_k$ Loss, and it exhibits the best performance. Experiments on synthetic datasets demonstrate that models trained with $\mathrm{ST}_k$ Loss most closely approximate the theoretical decision boundary. On benchmarks of imbalanced binary classification, models trained by $\mathrm{ST}_k$ Loss exhibit the lowest average misclassification rate and the lowest standard deviation. On regression datasets, models trained with $\mathrm{ST}_k$ Loss exhibit the lowest (RMSE). Experiments on large real-world datasets demonstrate that $\mathrm{ST}_k$ Loss, as an aggregating trick for individual losses, is a scalable technique that improves accuracy on long-tailed benchmarks. With the help of $\mathrm{ST}_k$ Loss, we surpass the state-of-the-art (SOTA) on both the CIFAR-100-LT and Places-LT leaderboards.

## 1.1 Related Work

In current research, although there are many studies on individual loss, the general characteristics of aggregate loss are often overlooked. In the existing machine learning literature, a related line of work is the data subset selection problem [Wei et al., 2015], which aims to select a subset from a large training dataset for model training while minimizing average loss. Curriculum learning [Bengio et al., 2009] and self-paced [Kumar et al., 2010] learning are two recent learning schemes. They organized the training process into several iterative stages, gradually including training data from easy to difficult to learn, where the difficulty level is measured by individual loss. Therefore, each training session in these methods corresponds to the average aggregate loss in the selected subset. The difficulties encountered by the Average Loss when dealing with imbalanced data, as discussed in [Shalev-Shwartz and Wexler, 2016, Huang et al., 2020], prompted the exploration of more robust aggregate losses. Among these, Lyu et al. [2020] introduced the $\mathrm{AT}_k$ loss, which averages the $k$ largest individual losses, and exhibits advanced performance on imbalanced datasets.

Many classification tasks in the real world have inherent label confusion, as mentioned in Berrada et al. [2018]. This confusion may arise from various factors, such as incorrect labels, incomplete annotations, or some fundamental ambiguities that even confuse the true labels for human experts. Therefore, some works proposed the concept of Top-k Learning in the field of image classification to address the issues of multiple semantics and semantic confusion in images [Lapin et al., 2017].

Berrada et al. [2018] proposed a method to partially smooth the Top-k Learning loss function, but did not completely solve the sorting problem in the loss function and introduced a sorting computation of $C_n^k$. Petersen et al. [2022] proposed a Split Selection Network (SSN) based on sorting networks, which made the Top-k process differentiable and achieved the state of the art on ImageNet-1K at that time. However, the computation required by this method is cumbersome and multilayered. Sorting networks are similar to algorithms like QuickSort with time complexities of $\mathcal{O}(n \log n)$.

## 1.2 Notational Conventions

Let $\mathbb{N}^n$ denotes the set $\{1, ..., n\}$ and $\mathbb{I}_{\{a\}}$ denotes the indicator function (which is 1 when the proposition $a$ is true and 0 otherwise). Thus, the sign function can be defined as: $sign(x) = \mathbb{I}_{\{x>0\}} - \mathbb{I}_{\{x<0\}}$. The Hinge function can be defined as: $[x]_+ = \max\{0, x\}$. We use $\|x\|_1$, $\|x\|_2$, and $\|x\|_\infty$ to represent the $\ell_1$, $\ell_2$, and $\ell_\infty$ norms of $x$, respectively. For the set $L = \ell_1, \ell_2, ..., \ell_n$, $\ell_{[k]}$ denotes the $k$-th largest element, so we have $\ell_{[1]} \geq \ell_{[2]} \geq ... \geq \ell_{[n]}$. In supervised learning problems, our training set typically contains an input set and a target set, the input set coming from the input domain $\mathcal{X}$, and the target set from the target domain $\mathcal{Y}$, and we use their joint domain $\mathcal{Z} = \mathcal{X} \times \mathcal{Y}$ to represent the range of the dataset. The training set $S = z_1, z_2, ..., z_n$ is a subset of $\mathcal{Z}$, where $z_i = (x_i, y_i)$. Our task is to find a predictor $f : \mathcal{X} \to \mathcal{Y}$ from the function family $\mathcal{H}$ that can predict the corresponding target $y$ based on the new input $x$. To evaluate the effect of the predictor, we need to introduce an individual loss function $\ell : \mathcal{Y} \times \mathcal{Y} \to \mathbb{R}^+$, where $\ell = \ell(f(x), y)$ usually reflects some distance between the prediction $\hat{y} = f(x)$ and the true value $y$. The training

process of the predictor can be described as using the gradient descent algorithm to optimize an objective function (usually minimizing the loss function). The objective function can generally be written as $\mathcal{L}(f, S) + \Omega(f)$, where $\mathcal{L}(f, S)$ is the aggregated individual loss function, and $\Omega(f)$ is a regularization term ($\ell_1$ or $\ell_2$ regularization term). The loss function $\mathcal{L}(f, S)$ is usually Average Loss, that is, $\mathcal{L}_{avg}(f, S) = \frac{1}{n} \sum_{i=1}^{n} \ell(f(x_i), y_i)$. However, recent works, have shown some drawbacks of the average loss in adapting to imbalanced data distributions [Shalev-Shwartz and Wexler, 2016], and explored choices other than the average loss for the aggregate loss formed from individual losses, e.g., the maximum (aggregate) loss, $\mathcal{L}_{max}(f, S) = \max_{i \in \mathbb{N}_n} \ell(f(x_i), y_i)$.

## 2 $\text{ST}_k$ Architecture

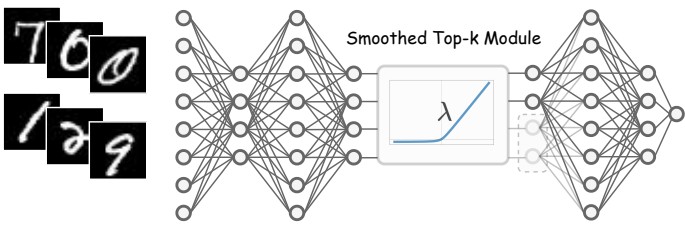

Figure 1: $\text{ST}_k$ Architecture. For any layer of neurons in a neural network, to solve the Top-k problem for its weights, insert an $\text{ST}_k$ Module. The trainable parameter $\lambda$ will gradually approximate the $k$-th largest element during the optimization process. And this $\lambda$ can be used to filter neurons.

Suppose we have a set of elements $\{e_i\}_{i=1}^{n}$, then the Top-k problem can be describe as:

1. find the $k$-th largest element $e_{[k]}$;

2. find the sum of top-k largest elements $\sum_{i=1}^{k} e_{[k]}$.

This process can certainly be achieved through conventional sorting and summation. However, in the worst-case scenario, the cost of solving the Top-k problem can reach $\mathcal{O}(n^2)$. Ogryczak and Tamir [2003] proposed an equivalent optimization form to solve the Top-k problem and proved its linear convergence.

$$\sum_{i=1}^{k} e_{[k]} = \min_{\lambda \geq 0} \left\{ \sum_{i=1}^{n} [e_i - \lambda]_+ + k\lambda \right\}, \tag{1}$$

However, this surrogate objective function suffers from a non-differentiable point at $e_i = \lambda$, which makes it challenging to optimize. The key to solving this problem lies in designing a function approximating $[\cdot]_+$, which can be regarded as a rectified linear unit (ReLU) function. Here, we introduce the Smoothed ReLU (SReLU).

$$\text{SReLU(x)} = \frac{1}{2} \left[ x + \delta \left( \sqrt{\frac{x^2}{\delta^2} + 1} - 1 \right) \right], \tag{2}$$

where $\delta$ is a hyperparameter (usually we set $\delta = 0.01$). It can be observed from Figure 2 that as $\delta$ decreases, SReLU increasingly approximates the ReLU function. In fact, SReLU converges uniformly to ReLU as $\delta \to 0^+$; a detailed proof of this uniform convergence is provided in Proposition 1 of Appendix A.1. With the help of SReLU, the objective function (1) can be smoothed as:

$$\sum_{i=1}^{k} e_{[k]} \approx \min_{\lambda \geq 0} \left\{ \frac{1}{2} \sum_{i=1}^{n} \left[ (e_i - \lambda) + \delta \left( \sqrt{\frac{(e_i - \lambda)^2}{\delta^2} + 1} - 1 \right) \right] + k\lambda \right\}, \tag{3}$$

the optimal $\lambda^* = e_{[k]}$ which is the $k$-th largest element. In the following section, we will introduce an application scenario for $\text{ST}_k$.

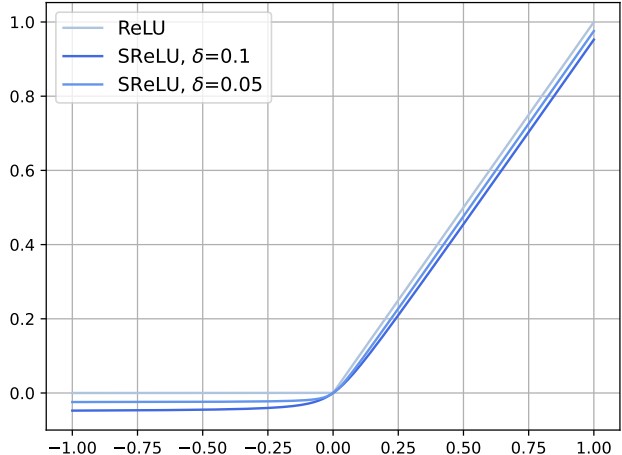

Figure 2: ReLU and SReLU with various smoothing coefficients $\delta$.

## 3 From AT$_k$ Loss to ST$_k$ Loss

In the training process of a neural network, we first choose a form of individual loss (e.g., logistic loss, hinge loss, mean square loss, or cross-entropy loss). Then, we aggregate all individual losses to calculate their average, which is the most common aggregate loss function: Average Loss. The Average Loss is widely used in a myriad of deep learning tasks. This widespread application stems from the robust theoretical foundations [Bartlett et al., 2006, De Vito et al., 2005]. However, Average Loss tends to overfit the training data, especially on imbalanced datasets [Shalev-Shwartz and Wexler, 2016, Huang et al., 2020]. This has inspired the motivation to find other forms of aggregate loss, such as the maximum value among individual losses (referred to as Maximum Loss). The Average Top-k (AT$_k$) Loss was introduced by Lyu et al. [2020]:

$$\mathcal{L}_{at-k} = \frac{1}{k}\sum_{i=1}^{k}\ell_{[k]}, \tag{4}$$

which represents the average of the largest $k$ individual losses. According to the derivation in [Ogryczak and Tamir, 2003], this ranking loss can be written in the following equivalent form:

$$\mathcal{L}_{mat-k} = \frac{1}{n}\sum_{i=1}^{n}[\ell_i - \lambda]_+ + \frac{k}{n}\lambda. \tag{5}$$

With the help of the ST$_k$ Module, $\mathcal{L}_{mat-k}(f, S)$ can be reconstructed as

$$\mathcal{L}_{st-k} = \frac{1}{2n}\sum_{i=1}^{n}\left[(\ell_i - \lambda) + \delta\left(\sqrt{\frac{(\ell_i - \lambda)^2}{\delta^2} + 1} - 1\right)\right] + \frac{k}{n}\lambda, \tag{6}$$

where $\ell_i = \ell(f_{w_{\text{model}}}(x_i), y_i)$ is the individual loss of sample $i$, and $w_{\text{model}}$ represents the set of parameters of the predictor $f$. It is easy to verify that $\lim_{\delta \to 0^+} \mathcal{L}_{st-k} = \mathcal{L}_{mat-k}$, given Proposition

1 in Appendix A.1.

$$\left[\frac{k\lambda}{n} + \frac{1}{n}\sum_{i=1}^{n}[\ell_i - \lambda]_+\right] - \left[\frac{k\lambda}{n} + \frac{1}{n}\sum_{i=1}^{n}\frac{1}{2}\left[(\ell_i - \lambda) + \delta\left(\sqrt{\frac{(\ell_i - \lambda)^2}{\delta^2} + 1} - 1\right)\right]\right]$$

$$= \sum_{i=1}^{n}\frac{1}{2n}\left[\sqrt{(\ell_i - \lambda)^2} - \sqrt{(\ell_i - \lambda)^2 + \delta^2} + \delta\right]$$

$$= \sum_{i=1}^{n}\frac{1}{2n}\left[\delta - \frac{\delta^2}{\sqrt{(\ell_i - \lambda)^2} + \sqrt{(\ell_i - \lambda)^2 + \delta^2}}\right]$$

$$= \sum_{i=1}^{n}\frac{1}{2n}\left[\delta\left(1 - \frac{1}{\sqrt{\frac{(\ell_i - \lambda)^2}{\delta^2}} + \sqrt{\frac{(\ell_i - \lambda)^2}{\delta^2} + 1}}\right)\right]$$

$$< \frac{\delta}{2}$$

The approximation error between the smoothed loss function and the original loss function can be uniformly bounded by $\delta/2, \forall \ell$.

When $\ell$ is convex, Equation (6) exhibits joint convexity with respect to the parameters $(w_{\text{model}}, \lambda)$, making the problem a special case of the non-linear multiple choice knapsack problem [Zemel, 1984], which has at most $q = 2$ roots. These roots can be found in constant time, allowing the problem to be solved in $\mathcal{O}(n \cdot \ln q) = \mathcal{O}(n)$ time when $q$ is fixed [Megiddo, 1984]. Therefore, it can be iteratively updated using relatively simple algorithms. For example, in the case of batch learning, the block coordinate descent (BCD) method [Nocedal and Wright, 1999] can be employed, where $w_{\text{model}}$ and $\lambda$ are updated alternately after initialization.

BCD-ST$_k$:

$$\lambda^{(t+1)} \leftarrow \underset{\lambda}{\text{argmin}}\, \mathcal{L}_{st-k};$$
$$w^{(t+1)} \leftarrow \underset{w}{\text{argmin}}\, \mathcal{L}_{st-k}. \tag{7}$$

The convergence of the above coordinate descent algorithms can be found in Luo and Tseng [1992], Saha and Tewari [2013], Tseng [2001].

Furthermore, empirical evidence suggests that we do not need to spend extra time optimizing $\lambda$ separately, incorporating $\lambda$ into the computational graph of $w_{\text{model}}$ for unified optimization using Stochastic Gradient Descent (SGD) [Bottou and Bousquet, 2008, Shamir, 2011, Srebro and Tewari, 2010], performance improvements can still be achieved.

SGD-ST$_k$:

$$\lambda^{(t+1)} \leftarrow \lambda^{(t)} - \eta \cdot \partial_\lambda \mathcal{L}_{st-k};$$
$$w^{(t+1)} \leftarrow w^{(t)} - \eta \cdot \nabla_w \mathcal{L}_{st-k}.$$

where $\eta_t$ is the size of the update step, and when $\eta_t \sim \frac{1}{\sqrt{t'}}$, the stochastic gradient descent method can ensure convergence to a local minimum of Equation (6) [Shamir, 2011, Srebro and Tewari, 2010]. By eliminating points where the gradients are discontinuous, the training process becomes more stable, and converges faster, as experimentally demonstrated by the standard deviations reported in Tables 3, 4, and 9, and the time cost reported in Table 2.

## 4 Synthetic Experiments

### 4.1 Time Cost

We first perform experiments to compare the time costs of two standard sorting algorithms, AT$_k$, and ST$_k$, in calculating the ranking average. The experimental setup involves finding the Top-k (k=5) sum from 10,000 standard normally distributed samples. For both AT$_k$ and ST$_k$, we iterate until the error is less than $10^{-2}$. For each algorithm, we run 50 experiments and record the average time taken.

| Algorithm | Complexity | Average Time(s) |
|---|---|---|
| BubbleSort | $\mathcal{O}(n^2)$ | $20.42196 \pm 3.7015$ |
| HeapSort | $\mathcal{O}(n\log(n))$ | $0.1243 \pm 0.0446$ |
| $\text{AT}_k$ | $\mathcal{O}(n+k)$ | $0.2167 \pm 0.1528$ |
| $\text{ST}_k$ (Ours) | $\mathcal{O}(n+k)$ | $\mathbf{0.0127 \pm 0.0020}$ |

Table 1: Performance comparison of different sorting algorithms and our method.

As shown in Table 1, $\text{ST}_k$ indeed demonstrates linear time complexity while also exhibiting a stable optimization process.

## 4.2 Gaussian Distributed Dataset

To illustrate the capability of $\text{ST}_k$ Loss in approximating the ideal decision boundary, we design a Gaussian-distributed dataset. The dataset is generated by a pre-set covariance matrix $\boldsymbol{\Sigma} \in \mathbb{R}^{d \times d}$, and mean vectors $\boldsymbol{\mu}_1, \boldsymbol{\mu}_2 \in \mathbb{R}^d$ corresponding to two categories. We set $d = 200$, $\boldsymbol{\Sigma}_{jk} = 0.25^{0.5+|j-k|}$, $\boldsymbol{\mu}_1 = \mathbf{0}_d$ and $\boldsymbol{\mu}_2 = (1, 1, ..., 1, 0, 0, ..., 0)^\top$, which has 10 ones. For the predictor, we use a Logistic Regression (LR) model $f(\boldsymbol{x}) = \text{sigmoid}(\boldsymbol{\omega}^\top \boldsymbol{x} + b)$. The detailed derivation process of the decision boundary can be found in the Appendix A.2. Here, we present the conclusion directly. For two normal populations with a given covariance matrix $\boldsymbol{\Sigma}$, and means $\boldsymbol{\mu}_1$ and $\boldsymbol{\mu}_2$, the LR model has a theoretical decision boundary:

$$\boldsymbol{\omega}^* = (\boldsymbol{\mu}_1 - \boldsymbol{\mu}_0)^\top \boldsymbol{\Sigma}^{-1}; \;\; b^* = \frac{1}{2}\left[\boldsymbol{\mu}_0^\top \boldsymbol{\Sigma}^{-1}\boldsymbol{\mu}_0 + \boldsymbol{\mu}_1^\top \boldsymbol{\Sigma}^{-1}\boldsymbol{\mu}_1\right] + \ln\left[\frac{\Pr(Y=1)}{\Pr(Y=0)}\right]. \tag{8}$$

Figure 3 provide an example on 2D plane, we set $\boldsymbol{\Sigma}_{(j,k)} = 0.8^{|j-k|}$, $\boldsymbol{\mu}_1 = [0,0]^\top$, $\boldsymbol{\mu}_2 = [2,2]^\top$, thus according to our derivation above, $\boldsymbol{\omega}^* = [1.111, 1.111]^\top$, $b^* = -0.836$, where the black dashed line is the theoretical boundary.

We use $\text{ParaF}_1$ [Tian and Gu, 2017] to measure the overlap of estimated supports and true supports:

$$\text{ParaF}_1 = 2 \cdot \frac{\text{precision} \cdot \text{recall}}{\text{precision} + \text{recall}},$$

where precision $= \frac{|\text{support}(\hat{\boldsymbol{\omega}}) \cap \text{support}(\boldsymbol{\omega}^*)|}{|\text{support}(\hat{\boldsymbol{\omega}})|}$, recall $= \frac{|\text{support}(\hat{\boldsymbol{\omega}}) \cap \text{support}(\boldsymbol{\omega}^*)|}{|\text{support}(\boldsymbol{\omega}^*)|}$, where $\hat{\boldsymbol{\omega}}$ is the estimated value of the parameters obtained by the predictor, and $\boldsymbol{\omega}^*$ is the theoretical value of the parameters mentioned above. The operator $|\cdot|$ is used to find the cardinality of the set, which is the number of elements, and $\text{support}(\cdot)$ is the support set of the vector, which refers to the set of indices of its non-zero elements.

Other basic settings are as follows. The loss function we use is the binary cross-entropy loss. To obtain a sparse solution, we add an $\ell_1$ regularization term. The number of samples is 10,000 for the training set and 2,500 each for the validation set and the test set. We implement early stopping of the iteration based on the accuracy curve in the validation set; that is, we break when the increase in validation accuracy within 200 steps is less than $10^{-4}$. For each meta-experiment, we repeat it 50 times and take the average. Now we conduct two groups of experiments as follows.

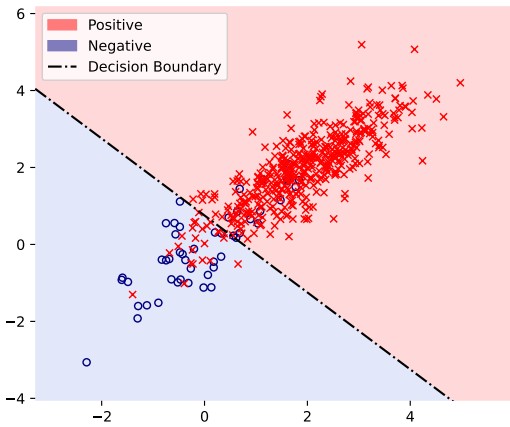

Figure 3: A Synthetic Example on 2D-Plain.

**Aggregate Losses and ReLU Varients.** In these experiments, we set the ratio of positive to negative samples at 8:2 for the training set, while the validation set and the test set remain 1:1. We also compare SReLU with other variants of ReLU. The smoothing coefficient for SReLU is set to 0.01, and the settings for the other ReLU variants remain at their defaults. We use Adam as our optimizer,

setting the batch size to 512, while keeping the other hyperparameters as default. According to Equation (8), we could compute the theoretical value of $\omega^*$, which is a vector with only the first 10 elements non-zero. **Sensitive Analysis.** In these experiments, we only adjust the positive-negative ratio and retain the rest of the settings from the previous experiment.

| Aggregate Loss | Average | Maximum | $AT_k$ | $MAT_k$ | $ST_k$ | | | |
| --- | --- | --- | --- | --- | --- | --- | --- | --- |
| | | | | | ELU | SoftPlus | Leaky-ReLU | SReLU (Ours) |
| Accuracy (%) | 72.864 | 65.448 | 73.013 | 73.180 | 72.988 | 70.968 | 73.864 | **76.104** |
| $ParaF_1$-Score | 0.1904 | 0.0909 | 0.1986 | 0.2063 | 0.1925 | 0.1592 | 0.2441 | **0.3446** |
| Time (s) | 16.525 | 6.145 | 19.271 | 15.774 | 15.572 | 16.233 | 15.525 | 15.436 |

Table 2: Accuracy and $ParaF_1$-Score on the synthetic dataset.

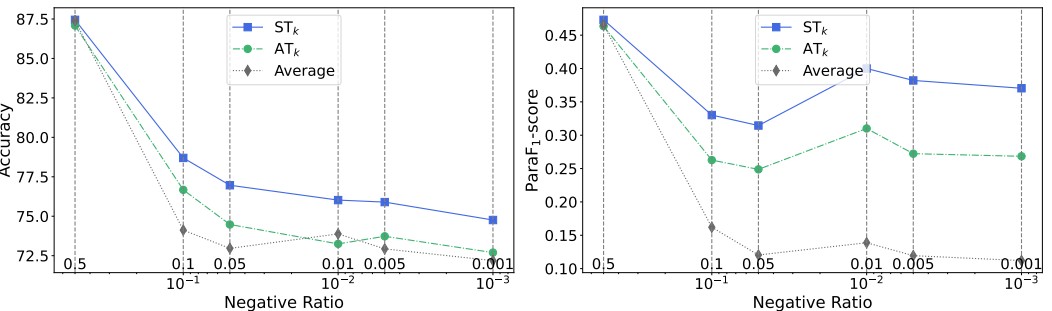

Figure 4: Accuracy and $ParaF_1$-Score vs Negative Sample Ratio.

Table 2 presents the Accuracy and $ParaF_1$ score achieved when different aggregate loss functions are combined with the LR model and the cross-entropy loss, trained to convergence. With the help of SReLU, $ST_k$ Loss achieves performance that surpasses that of all other aggregate losses. Due to its property of being a real uniformly convergent approximation (not just similar in shape), SReLU outperforms other ReLU variants in this scenario. Figure 4 shows the relationship between accuracy, $ParaF_1$-score, and negative sample ratio (negative ratio). $ST_k$ is less sensitive to the imbalance ratio,

## 5 Real World Applications

### 5.1 Binary Classification

We select binary classification datasets from the KEEL[2] and UCI[3] databases; see Table 9.

| Dataset | Average | Maximum | $AT_k$ | $MAT_k$ | $ST_k$(Ours) |
| --- | --- | --- | --- | --- | --- |
| appendicitis | 13.778±6.601 | 32.815±11.306 | 13.630±6.400 | 14.667±7.114 | **13.406±5.491** |
| australian | 13.549±1.025 | 46.89±8.779 | 14.428±1.056 | 14.324±1.058 | **13.358±0.982** |
| german | 26.328±2.069 | 45.832±5.465 | 26.808±2.347 | 26.392±2.431 | **25.561±1.988** |
| phoneme | 20.823±2.801 | 45.83±10.819 | 21.420±2.732 | 17.828±2.709 | **16.427±2.494** |
| spambase | 6.972±1.519 | 45.777±19.418 | 6.955±1.535 | 6.687±1.577 | **6.610±1.333** |
| titanic | 22.613±1.226 | 48.065±11.338 | 22.468±1.441 | 22.211±1.095 | **21.801±0.911** |
| wisconsin | 3.275±0.814 | 34.468±9.306 | 3.205±0.773 | 3.046±0.570 | **2.936±0.665** |

Table 3: Misclassification Rate(%) and Standard Derivation of Various Aggregate Losses Combined with Individual Logistic Loss.

Next, we tested the performance of different forms of aggregate loss on binary classification benchmarks. The individual loss function can generally be chosen as Logistic or Hinge, which are defined

---

[2]http://sci2s.ugr.es/keel/datasets.php
[3]https://archive.ics.uci.edu/ml/datasets.html

| Dataset | Average | Maximum | $AT_k$ | $MAT_k$ | $ST_k$(Ours) |
|---------|---------|---------|--------|---------|--------------|
| appendicitis | 15.852±8.082 | 28.000±12.443 | 16.148±7.436 | 15.481±5.577 | **14.667±5.736** |
| wisconsin | 3.240±1.014 | 12.000±10.760 | 3.158±1.085 | 2.936±1.212 | **2.889±1.069** |
| australian | **13.225±2.351** | 36.347±9.576 | 13.896±2.671 | 14.243±2.343 | 13.746±1.966 |
| german | 26.912±2.694 | 43.864±12.432 | 26.776±3.010 | 26.232±2.190 | **25.928±2.973** |
| titanic | 22.152±1.862 | 47.691±18.858 | 22.316±1.490 | 22.116±1.567 | **21.966±1.289** |
| phoneme | 21.543±1.358 | 44.656±16.805 | 20.733±1.093 | 17.335±1.125 | **17.152±0.955** |
| spambase | 7.463±0.742 | 33.986±8.955 | 7.197±0.695 | 6.645±0.677 | **6.342±0.589** |

Table 4: Misclassification Rate (%) and Standard Derivation of Various Aggregate Losses Combined with Individual Hinge Loss.

as follows:

$$\text{Logistic} : \ell(f(x), y) = \log(1 + \exp(-yf(x)));$$
$$\text{Hinge} : \ell(f(x), y) = [1 - yf(x)]_+.$$

The prediction model is a two-layer MLP with 10 nodes in the hidden layer, activated by the ReLU function between the two fully connected layers. To increase the stability of the training process, we added an $\ell_2$ regularization term to the loss function $\Omega(w) = \frac{1}{2C}\|w\|_2^2$. We divided the datasets into training, validation, and test sets in a 0.5 : 0.25 : 0.25 ratio. Nowadays, SGD is generally replaced by the mini-batch method instead of stochastic gradient descent with a single sample point, which is faster and more robust. To accommodate datasets of varying sizes in the article, we set the batch size to 16.

The hyperparameters in the experiment include $k$ in $MAT_k$ and $AT_k$, the coefficient of the regularization term $C$, the initial learning rate $\eta$, and the smoothing coefficient $\delta$. These hyperparameters will be selected based on their convergence accuracy performance on the validation set (for each combination of hyperparameters, we repeat the experiment fifty times and take the average of their prediction accuracy on the validation set as the basis for parameter selection). The search spaces for several hyperparameters are as follows: $k \in \{1\} \cup [0.1 : 0.1 : 1]$; $C \in \{10^0, 10^1, 10^2, 10^3, 10^4, 10^5\}$; $\eta \in \{0.1, 0.05, 0, 01, 0.005, 0.001\}$; $\delta \in \{0.1, 0.01, 0.001, 0.0001\}$.

Since traditional gradient descent is too sensitive to the choice of step size, which is not conducive to our pure comparison of the convergence speed and accuracy of the loss function, we use the AdaGrad algorithm to iteratively update the learning rate. During the learning process, to avoid overfitting the model, we record the accuracy of the MLP predictor on the validation set after each iteration, and perform an early stop when the accuracy does not increase (the increase in the accuracy of the predictor on the validation set is less than $10^{-6}$ after 50 steps) and roll back the model to the checkpoint with the highest accuracy on the validation set during the entire training process; otherwise, we continue training until convergence. Tables 3 and 4, respectively, show the average probability of misclassification (%) of the models trained to convergence in 50 experiments in the test set under individual Logistic and Hinge loss (the standard deviation of the 50 experiments is listed in parentheses).

In the results shown in Tables 3 and 4. Almost all the lowest misclassification rates appear in the $ST_k$ trained models. Furthermore, models trained with $ST_k$ Loss exhibit the lowest standard deviation, which to some extent demonstrates the robustness of the training process.

## 5.2 Long-Tailed Classification

The interest in long-tailed classification tasks has increased with the advent of large vision-language models such as contrastive language-image pre-training (CLIP) Radford et al. [2021]. Long-tailed versions of recognized datasets were built by the community. Using the Pareto distribution ($\alpha = 6$), the ImageNet-1K and Places datasets can be sampled to create ImageNet-LT and Places-LT datasets, as described in [Liu et al., 2019]. Consider two types of imbalance [Cui et al., 2019, Buda et al., 2018], Cao et al. [2019] built CIFAR-10-LT ($\rho = 10$) and CIFAR-100-LT ($\rho = 100$), where $\rho = \max_i\{n_i\}/\min_i\{n_i\}$ represents the imbalance ratio, defined as the ratio between the sample sizes of the most frequent and least frequent class. Due to varying word frequencies, machine translation, or

more generally, text generating, can inherently be considered as long-tailed classification tasks. We choose the WMT2017[4] and IWSLT2014[5] datasets for the experiments.

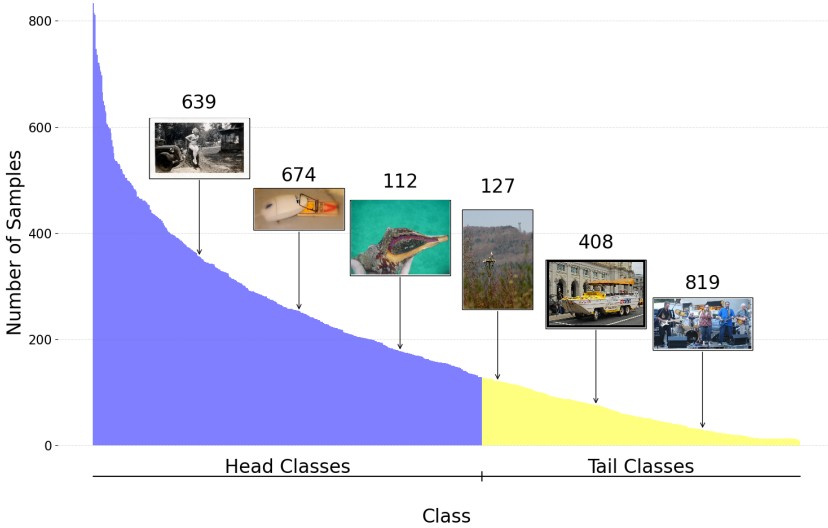

Figure 5: ImageNet-LT.

**Settings** For visual classification tasks, we use Parameter-Efficient Long-Tailed (PEL) Recognition [Shi et al., 2023], which primarily leverages information from a text encoder to adjust classification probabilities. The text encoder in PEL is derived from the pre-trained CLIP model, and the backbone is a Vision Transformer (ViT-Base) [Dosovitskiy et al., 2021] pre-trained in ImageNet-21K. We freeze the backbone and train the branch model of Parameter Efficient Fine-Tuning (PEFT) [Jia et al., 2022, Hu et al., 2022, Houlsby et al., 2019]. We simply replaced the Average aggregation of individual losses in the Logit Adjust (LA) [Menon et al., 2021] loss with $AT_k$ or $ST_k$. For $k$, we set it at $0.9 \times$ batch size, and $\delta = 0.01$. Then we report the accuracy of the balanced test sets. Visual classification experiments can be implemented on a single L20 GPU, with varying durations ranging from 20 to 200 minutes.

For translation tasks, we use the training workflows provided by OpenNMT[6] [Klein et al., 2018] and FAIR-Seq[7] [Ott et al., 2019], which are very easy to reproduce. Models are essentially different sizes of Transformers (see the footnotes). In the training process of language models, the actual batch size, calculated as the sequence length multiplied by the number of sequences, is often dynamic. Consequently, we set $k$/batch size $= 0.96$. We simply replaced the Average aggregation of individual losses in the Cross Entropy (CE) loss with $AT_k$ or $ST_k$. We calculate the bilingual evaluation understudy (BLEU) scores for the models. The BLEU for IWSLT2014-trained models are derived from their test set, while for the WMT2017-trained models, we evaluate their BLEU on the Newstest2016 test set. Training these Transformer-based models would take about 10 hours on a single RTX 4090.

**Overall Results** $ST_k$ Loss, as an aggregating trick for individual losses, is a scalable technique that improves accuracy on long-tailed benchmarks. The models trained by $ST_k$ Loss surpass SOTA on the CIFAR-100-LT and Places-LT leaderboards.

---

[4]https://www.statmt.org/wmt17/

[5]https://workshop2014.iwslt.org/

[6]https://github.com/OpenNMT/OpenNMT-py/blob/master/docs/source/examples/wmt17/Translation.md

[7]https://fairseq.readthedocs.io/en/latest/_modules/fairseq/models/transformer/transformer_legacy.html

|  | CIFAR-100-LT | ImageNet-LT | Place-LT | IWSLT2014 | WMT2017 |
|---|---|---|---|---|---|
| Average | $89.1^*$ | 78.3 | $52.2^*$ | 35.27 | 35.21 |
| $AT_k$ | 89.3 | 78.1 | 52.3 | 35.45 | 35.05 |
| $ST_k$ | **89.8** | **79.1** | **53.7** | **35.80** | **35.42** |

Table 5: Results on large long-tailed datasets are presented, where the first three columns indicate accuracy, and the last two columns show the BLEU scores. Values marked with an "*" represent the state-of-the-art (SOTA) on the leaderboard.

## 5.3 Regression Tasks

For descriptions of the four regression datasets, see Table 9. In regression tasks, the individual loss is generally the distance ($\ell_1$ or $\ell_2$ distance) between the predicted value and the ground truth. Next, we present the specific settings of the experiment. For the Housing, Abalone, and Cpusmall datasets, we normalize the output to between $[0, 1]$; for the Sinc dataset, we first randomly sample 1000 points $(x_i, y_i)$ from the function, where $x_i \in [-10, 10]$, then we use the Radial Basis Function (RBF) kernel to map $x_i$ into the kernel space. We select 10 RBF kernels, resulting in a 10-dimensional input $x = [k(x, c_1), ..., k(x, c_{10})]$, where $k(x, c_j) = \exp(-(x - c_j)^2)$. Furthermore, we add Gaussian noise $\mathcal{N}(0, 0.2^2)$ to the target output $y$.

| Datasets | Square Individual Loss | | | Absolute Individual Loss | | |
|---|---|---|---|---|---|---|
|  | Average | $MAT_k$ | $ST_k$ (Ours) | Average | $MAT_k$ | $ST_k$ (Ours) |
| Sinc | 0.1603±0.0065 | 0.1591±0.0068 | **0.1446±0.0070** | 0.1686±0.0064 | 0.1560±0.0067 | **0.1457±0.0043** |
| Housing | 0.0896±0.0083 | 0.0816±0.0100 | **0.0810±0.0096** | 0.0781±0.0099 | 0.0670±0.0083 | **0.0626±0.0089** |
| Abalone | 0.0739±0.0018 | 0.0739±0.0018 | **0.0523±0.0014** | 0.0740±0.0018 | 0.0733±0.0018 | **0.0517±0.0021** |
| Cpusmall | 0.0613±0.0012 | 0.0313±0.0005 | **0.0296±0.0005** | 0.0613±0.0012 | 0.0313±0.0005 | **0.0244±0.0006** |

Table 6: Comparison of Average RMSE and Standard Deviation for Different Aggregate Losses Combined with Square and Absolute Loss.

For individual losses, we choose the absolute value loss ($\ell_1$) and the squared loss ($\ell_2$). The prediction model is the same as above, a two-layer MLP with 10 nodes in the hidden layer, activated by the ReLU function between the two fully connected layers. As in binary classification experiments, we randomly divide the dataset into training, validation, and test sets, with hyperparameters chosen based on the performance in the validation set, repeated 50 times, and statistics on the test set are reported.

Tables 6 show the average RMSE and standard deviation of 50 experiments in regression datasets with different aggregate losses combined with absolute individual loss and squared individual loss (given the experience of the previous experiment, we no longer compare with Maximum Loss here), and $ST_k$ shows further performance improvement after smoothing.

## 6 Conclusion and Limitation

The proposed $ST_k$ Module can effectively solve the Top-k problem within neural networks, with no additional GPU memory or ranking time. Due to its fully differentiable nature, the training process are stable. By applying $ST_k$ Module to the Average Top-k Loss, we achieve significant improvements across numerous benchmarks.

The limitation of this study is that we have not yet evaluated the $ST_k$ Module for additional application scenarios. The effectiveness of the $ST_k$ Module is only demonstrated in smoothing $AT_k$ loss. We hope that future research will further explore the potential utility of the $ST_k$ Module.

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

# A Appendix: Proposition Proof

## A.1 Uniform Convergence

**[Proposition 1]** We have that SReLU uniformly converges to ReLU as $\delta \to 0^+$, that is, $\lim_{\delta \to 0^+} \sup_x |\text{SReLU}_\delta(x) - [x]_+| = 0$.

**Proof.**

$$\text{ReLU}(x) = [x]_+ = \frac{1}{2}(x + |x|);$$

$$\text{SReLU(x)} = \frac{1}{2}\left[x + \delta\left(\sqrt{\frac{x^2}{\delta^2} + 1} - 1\right)\right].$$

Note that $\text{D}(x) = 2(\text{ReLU}(x) - \text{SReLU}(x)) = |x| - \delta\left(\sqrt{\frac{x^2}{\delta^2} + 1} - 1\right)$, we have

$$\text{D}(x) = \sqrt{x^2} - \sqrt{x^2 + \delta^2} + \delta$$

$$= \delta - \frac{\delta^2}{\sqrt{x^2} + \sqrt{x^2 + \delta^2}}$$

$$= \delta\left(1 - \frac{1}{\sqrt{\frac{x^2}{\delta^2}} + \sqrt{\frac{x^2}{\delta^2} + 1}}\right).$$

For any $x$ and $\delta$, the form $0 < 1 - \frac{1}{\sqrt{\frac{x^2}{\delta^2}} + \sqrt{\frac{x^2}{\delta^2} + 1}} < 1$, therefore, for any $\epsilon > 0$, if $\delta < \epsilon$, then $\forall x$ we shall have $\text{D}(x) < \epsilon$.

## A.2 Decision Boundary

**[Proposition 2]** For two normal populations with known (denote as $\boldsymbol{\Sigma}$) and equal covariance matrices, and means $\boldsymbol{\mu}_1$ and $\boldsymbol{\mu}_2$, the LR model $f(\boldsymbol{x}) = \text{sigmoid}(\boldsymbol{\omega}^\top \boldsymbol{x} + b)$ has a theoretical decision boundary:

$$\boldsymbol{\omega}^* = (\boldsymbol{\mu}_1 - \boldsymbol{\mu}_0)^\top \boldsymbol{\Sigma}^{-1};$$

$$b^* = \frac{1}{2}\left[\boldsymbol{\mu}_0^\top \boldsymbol{\Sigma}^{-1} \boldsymbol{\mu}_0 + \boldsymbol{\mu}_1^\top \boldsymbol{\Sigma}^{-1} \boldsymbol{\mu}_1\right] + \ln\left[\frac{\Pr(Y=1)}{\Pr(Y=0)}\right].$$

**Proof.** Usually, we estimate the parameters $\boldsymbol{\omega}$ and $b$ by maximizing the likelihood function

$$\frac{1}{n}\sum_{i=1}^n \ln[\Pr(y_i|x_i)].$$

This is to estimate parameters in the "most data-fitting manner" when the distribution is unknown. Now we know that two classes are normally distributed, we even know their mean and covariance matrix.

In Logistic Regression

$$\Pr(Y = 1|\boldsymbol{x}) = \frac{1}{1 + e^{-(\boldsymbol{\omega}^\top \boldsymbol{x} + b)}};$$

$$\Pr(Y = 0|\boldsymbol{x}) = \frac{e^{-\boldsymbol{\omega}^\top \boldsymbol{x} + b}}{1 + e^{-(\boldsymbol{\omega}^\top \boldsymbol{x} + b)}}.$$

So we have

$$\ln\left[\frac{\Pr(Y=1|\boldsymbol{x})}{\Pr(Y=0|\boldsymbol{x})}\right] = \boldsymbol{\omega}^\top \boldsymbol{x} + b. \tag{9}$$

And by Bayes' theorem, we can derive

$$\ln\left[\frac{\Pr(Y=1|\boldsymbol{x})}{\Pr(Y=0|\boldsymbol{x})}\right] = \ln\left[\frac{\Pr(\boldsymbol{x}|Y=1)\Pr(Y=1)}{\Pr(\boldsymbol{x}|Y=0)\Pr(Y=0)}\right].$$

Here, since we inherently know the data is normally distributed, we have

$$\ln\left[\frac{\Pr(Y=1|\boldsymbol{x})}{\Pr(Y=0|\boldsymbol{x})}\right] = (\boldsymbol{\mu}_1 - \boldsymbol{\mu}_0)^\top \boldsymbol{\Sigma}^{-1}\boldsymbol{x} + \frac{1}{2}\left[\boldsymbol{\mu}_0^\top \boldsymbol{\Sigma}^{-1}\boldsymbol{\mu}_0 + \boldsymbol{\mu}_1^\top \boldsymbol{\Sigma}^{-1}\boldsymbol{\mu}_1\right] + \ln\left[\frac{\Pr(Y=1)}{\Pr(Y=0)}\right]. \tag{10}$$

Then, comparing Equation (10) with Equation (9), we'll have the answer.

### A.3 More Experiments

We conduct an ablation study on "how $ST_k$ complements other long-tailed learning algorithms." The backbone here is a Vision Transformer (ViT) pre-trained by MAE / CLIP [He et al., 2022, Radford et al., 2021]. "CS" represents cost-sensitive learning, and "PEL" refers to the method provided by [Shi et al., 2023]. The batch size was set to 2048 and the models were trained in 30,000 steps.

| Pretrained By | CS | PEL | $ST_k$ | ImageNet-LT | CIFAR-100-LT |
|---|---|---|---|---|---|
| MAE | – | – | – | 65.701 | 78.572 |
| MAE | ✓ | – | – | 69.884 | 82.135 |
| MAE | ✓ | – | ✓ | 70.140 | 83.048 |
| CLIP | – | ✓ | – | 78.296 | 89.103 |
| CLIP | – | ✓ | ✓ | 79.148 | 89.833 |

Table 7: Ablation Study on Long-Tailed Learning Algorithms.

The smoothing coefficient $\delta$, 0.01, is a grid search determined value for all datasets in Section 5.1 (see page 7) and was adopted in all other experiments. We conducted an ablation study on the five real-world datasets with all other settings remaining unchanged, except for $\delta$.

| $\delta$ | CIFAR-100-LT | ImageNet-LT | Place-LT | IWSLT2014 | WMT2017 |
|---|---|---|---|---|---|
| 0.1 | 88.78 | 76.11 | 51.06 | 33.12 | 32.07 |
| 0.05 | 89.44 | 78.96 | 53.80 | 34.82 | 35.28 |
| 0.01 | 89.83 | 79.09 | 53.69 | 35.80 | 35.42 |
| 0.005 | 89.80 | 79.16 | 53.75 | 35.67 | 35.25 |

Table 8: Sensitive Analysis of the Smoothing Coefficient $\delta$.

### A.4 Benchmarks Detailed Information

| | Regression Datasets | | | | Binary Classification Datasets | | | | | | |
|---|---|---|---|---|---|---|---|---|---|---|---|
| | Sinc | Housing | Abalone | Cpusmall | Appendicitis | Australian | German | Phoneme | Spambase | Titanic | Wisconsin |
| n | 1000 | 506 | 4177 | 8192 | 106 | 690 | 1000 | 5404 | 4601 | 2201 | 683 |
| d | 10 | 13 | 8 | 12 | 7 | 14 | 24 | 5 | 57 | 3 | 9 |
| c | – | – | – | – | 2 | 2 | 2 | 2 | 2 | 2 | 2 |

Table 9: Statistics of Benchmarks.

In Table 9, we provide the statistical information for two benchmark sets discussed in Section 5, where c, n, d are the number of classes, samples and features, respectively. The left part provides the information of four regression datasets in Section 5.3. The "Sinc" dataset is a synthetic dataset, sampled from the sinc function $y = \sin(x)/x$. The right part provides the information of seven binary classification datasets in Section 5.1.

