# OpenReview forum: "ST$_k$: A Scalable Module for Solving Top-k Problems"
_NeurIPS.cc/2024/Conference — NeurIPS 2024 poster_

### Official Review · Reviewer_SwHE · 2024-06-14

**Soundness:** 3
**Presentation:** 3
**Contribution:** 2
**Rating:** 6
**Confidence:** 3

**Summary:**

This paper addresses the Top-K problem by introducing a new loss function. Building on the Average Top-K Loss, the authors incorporate a smoothed ReLU function to create the $ST_k$ loss, which is fully differentiable. Through experiments on various datasets, they demonstrate the effectiveness of their approach.

**Strengths:**

1. The paper is clearly written and easy to follow.
2. The authors conduct extensive experiments to evaluate the proposed $ST_k$ loss, using both synthetic and real-world datasets.

**Weaknesses:**

1. The core weakness lies in the lack of significant innovation. The main technical contribution is the replacement of the ReLU-like operation in an existing optimization method for the Top-K problem with a smoothed ReLU function, controlled by a hyperparameter for smoothness. Similar concepts already exist in various ReLU function variants, as seen in related works such as [1,2].
2. The only large-scale experiment is a long-tailed classification task involving parameter-efficient fine-tuning of a CLIP model pretrained on ImageNet-21K. Given that this model already has substantial pre-trained knowledge, the results are less convincing. Evaluating the method by training a model from scratch on these tasks would better demonstrate its effectiveness in practical scenarios.
3. The paper does not include an ablation study on the hyperparameter $\delta$ to study its impact on the performance.

[1]. Biswas, Koushik, et al. "SMU: smooth activation function for deep networks using smoothing maximum technique." arXiv preprint arXiv:2111.04682 (2021).

[2]. Hendrycks, Dan, and Kevin Gimpel. "Gaussian error linear units (gelus)." arXiv preprint arXiv:1606.08415 (2016).

**Questions:**

1. In lines 20-21, the authors mention that "the cost of the ranking process can no longer be ignored." I am curious about how significant this ranking cost is in the context of the full forward pass of a deep learning model. Given that larger models require more computational resources for their intermediate layers, this ranking cost might be negligible in most cases. Additionally, there is no detailed comparison of time costs between the Average Top-K Loss $AT_k$ and the proposed $ST_k$ loss, apart from on the toy synthetic dataset.

**Limitations:**

Yes.

---

> ### Author Rebuttal · Authors · 2024-08-07
>
> Thank you for your thorough review and feedback. Below is a detailed point-by-point response addressing your main concerns and questions.
>
>
> **For your concerns:**
>
>
> > lack of significant innovation
>
> We hold a different view on this.
>
> * The sorting optimization algorithm proposed by [1] has been around for two decades and has not been widely applied, largely due to the instability caused by the non-differentiable nature of the optimization function.
> * This paper introduces a new SReLU that possesses point-wise convergence to the original ReLU, a feature not present in other shape-similar ReLU variants. Our experiments demonstrate that SReLU surpasses other ReLU variants at least in solving the Top-K problem.
> * We have adapted the improved version of [1] for training deep learning models, and our experiments have shown that the trainable parameter $\lambda$ and model parameters from [1] do not require a two-step optimization using BCD but can be integrated into the computation graph and optimized uniformly with SGD.
> * Utilizing ST$_k$, we refresh the state-of-the-art (SOTA) records for two long-tailed classification leaderboards without additional computational resources. Moreover, ST$_k$ is almost complementary to any existing long-tailed learning methods. This is likely to attract widespread attention from the relevant communities.
>
> [1] Ogryczak, Wlodzimierz, and Arie Tamir. "Minimizing the sum of the k largest functions in linear time." Information Processing Letters 85.3 (2003): 117-122.
>
> > Evaluating the method by training a model from scratch
>
> In this work, we trained models from scratch on two machine translation datasets, and the experimental results show that ST$_k$ improves model performance. We will consider including results from training ImageNet-LT from scratch,  in our final version.
>
>
> > An ablation study on hyperparameter $\delta$.
>
> * The smoothing coefficient $\delta$, 0.01 is a grid search determined value for all datasets in section 5.1 (see page 7, line 186) and was adopted in all the other experiments.
> * We believe that the ablation study is necessary, and here we provide an ablation study on the CIFAR-100-LT dataset. All other settings remain the same as those in the paper, with only $\delta$ being changed.
>
> | $\delta$   | 0.1 | 0.05 | 0.01 | 0.005 |
> |--------------|-------|--------|-------|--------|
> | Accuracy | 88.8 | 89.4 | 89.8 | 89.8 |
>
> * We will include extensive ablation studies in our final version, thanks again for the insightful suggestion.
>
>
> **For your question:**
>
> > Is the ranking cost negligible？
>
> We experiment with the time cost of a few common sorting algorithms, AT$_k$ and ST$k$ to calculate the ranking average. The specific scheme of the experiment is to find the Top-k (k=5) sum from 10,000 normally distributed samples. For AT$_k$ and ST$_k$, we iterate until the error is less than $10^{-2}$, and for each algorithm, we conduct 50 experiments and record the average time taken in the third column of the Table.
>
> | Algorithm    | Complexity         | Average Time(s) |
> |--------------|--------------------|-----------------|
> | BubbleSort | $\mathcal{O}(n^2)$ | 20.42196$\pm$3.7015        |
> | HeapSort    | $\mathcal{O}(n\log(n))$ | 0.1243$\pm$0.0446   |
> | AT$_k$        | $\mathcal{O}(n+k)$ | 0.2167$\pm$0.1528     |
> | ST$_k$(Ours)| $\mathcal{O}(n+k)$ | **0.0127**$\pm$0.0020     |
>
> In experiments, ST$_k$ is the fastest algorithm to solve the ranking average value, on the other hand, AT$_k$ lacks robustness in training due to the gradient discontinuity.
>
> In addition, we conduct time cost experiment by training a Large Language Model. On a single H800, we tune a Llama-8B using LoRA method. After iterating 1000 steps, the average time cost is 1.9824s per step, while the performing the quick sort on the individual losses cost 0.0317s per step (# of tokens per batch = 2048).

---

> ### Author Response · Authors · 2024-08-11
>
> Thanks again for reviewing our work. As the author-reviewer discussion phase wraps up, could you let us know if our responses have addressed your concerns? If so, would you reconsider your rating? If you still have concerns, please share so we can address them.

---

> ### Comment · Reviewer_SwHE · 2024-08-12
> **Response**
>
> Thanks for the authors' response. Some of my concerns have been addressed in the rebuttal. However, after reviewing comments from other reviewers, such as reviewer uCvC, I also share curiosity about the theoretical support behind the proposed method, which has not been directly addressed in the rebuttal. The SReLU approach, in particular, seems to be an empirical result without clear theoretical guidance.

---

> ### Author Response · Authors · 2024-08-12
>
> Thanks for raising this issue, and we are glad to address this for all reviewers.
>
> **1.** The motivation here is that the discontinuity has affected the stability of the parameter $\lambda$ optimization process.
> Therefore, we need to find a smooth objective function to approximate:
>
> $\min_\lambda \frac{k \lambda}{n} + \frac{1}{n}\sum_{i=1}^n [\ell_i - \lambda]_+$
>
> **2.** Other ReLU-variants cannot sufficiently approximate ReLU as they are only similar in shape. Therefore, we design SReLU and proved that the approximation error can be bounded by $\delta / 2$ (see page 12, A.1).
>
>   $ \left[ \frac{k \lambda}{n} + \frac{1}{n}\sum_{i=1}^n [\ell_i - \lambda]_+ \right] $
>
>   $- \left[ \frac{k \lambda}{n} + \frac{1}{n}\sum_{i=1}^n \frac{1}{2}\left[(\ell_i-\lambda) + \delta \left( \sqrt{\frac{(\ell_i-\lambda)^2}{\delta^2}+1}-1 \right)\right] \right]$
>
>   $= \sum_{i=1}^n \frac{1}{2n}  \left[ \sqrt{(\ell_i-\lambda)^2} - \sqrt{(\ell_i-\lambda)^2 + \delta^2} + \delta \right]$
>
>   $= \sum_{i=1}^n \frac{1}{2n}  \left[ \delta - \frac{\delta^2}{\sqrt{(\ell_i-\lambda)^2} + \sqrt{(\ell_i-\lambda)^2 + \delta^2}} \right]$
>
>   $=  \sum_{i=1}^n \frac{1}{2n}  \left[ \delta \left( 1 - \frac{1}{\sqrt{\frac{(\ell_i-\lambda)^2}{\delta^2}} + \sqrt{\frac{(\ell_i-\lambda)^2}{\delta^2} + 1}} \right) \right] < \frac{\delta}{2}$
>
> **3.** The smoothed objective function shows jointly convex w.r.t $(w_{model}, \lambda)$, making ST$_k$ a special case of the non-linear multiple choice knapsack problems, with at most $q=2$ roots, which can be found in constant time. Thus, the problem can be solved in $\mathcal{O}(n*lnq)=\mathcal{O}(n)$ time as $q$ being fixed [1] (section 4, "non-linear case").
>
> [1] Zemel E. An O (n) algorithm for the linear multiple choice knapsack problem and related problems[J]. Information processing letters, 1984, 18(3): 123-128.
>
> [2] N. Megiddo, Linear programming in linear time when the dimension is fixed, J. ACM 31 (1984) 114–127.
>
> [3] Ogryczak, Wlodzimierz, and Arie Tamir. "Minimizing the sum of the k largest functions in linear time." Information Processing Letters 85.3 (2003): 117-122.
>
> **4.** Experiments have shown that, empirically, we don't even need to spend additional time optimizing $\lambda$ separately. Instead, by optimizing it synchronously with the model parameters $w_{model}$, we can still achieve performance improvements.
>
> Hope these explanations address your concerns. If you have any confusion, please feel free to reach out to us anytime.

---

> > ### Comment · Reviewer_SwHE · 2024-08-12
> >
> > Thank you for addressing my concerns about the theoretical support. I would like to increase my final rating.

---

> > > ### Author Response · Authors · 2024-08-12
> > >
> > > We appreciate your recommendation. Best wishes.

---

### Official Review · Reviewer_46Ey · 2024-07-10

**Soundness:** 2
**Presentation:** 3
**Contribution:** 2
**Rating:** 4
**Confidence:** 4

**Summary:**

This paper proposes a differentiable module for solving the top-k problem. Specifically, the paper proposes to approximate the hinge function with a new differentiable function.
Experiments on binary classification, long-tailed classification, and regression tasks with $ST_k$ loss show some improvements over baselines, like Average loss, and $AT_k$ loss.

**Strengths:**

(1) The paper is written clearly and easy to follow.
(2) The proposed new differentiable function in Eq.(2) can approximate the hinge function.
(3) The proposed method is flexible and is expected to be used in many scenarioes.

**Weaknesses:**

(1) For the implementation of $AT_k$, is it implemented by a sort algorithm or just Eq. (5)?
     From my point of view, the performance with $ST_k$ loss should be bounded by the performance with $AT_k$ loss.
     However, Tables 1,4 show that $ST_k$ outperforms $AT_k$.
     Why is $ST_k$ loss better than $AT_k$ loss?

(2) The proposed $ST_k$ loss is flexible and can be directly deployed on the traditional training paradigm.
     It would be better to validate the effectiveness of $ST_k$ on large-scale data, like ImageNet, and iNaturalist.

(3) The authors claim that the average loss is easily overfitted on imbalanced data. With $ST_k$ loss and CLIP models, it achieves the state-of-the-art on ImageNet-LT and Places-LT.
     Is the $ST_k$ loss complementary to other state-of-the-art long-tailed learning methods, like GPaCo[2] and BCL[1]?
     [1] Balanced Contrastive Learning for Long-Tailed Visual Recognition. CVPR 2022.
     [2] Generalized Parametric Contrastive Learning. TPAMI 2023.

(4) The models trained with $ST_k$ loss could be optimized alternatively or simultaneously with SGD for model parameters and $\lambda$. The ablation study is encouraged to be included.

**Questions:**

(1) Explanation of why $ST_k$ outperforms $AT_k$.
(2) The experiments on large-scale data, like ImageNet.
(3) Is the $ST_k$ loss complementary to current state-of-the-art long-tailed learning methods.
(4) Ablation for the optimization manner of model parameters and $\lambda$.

**Limitations:**

Limitations are discussed.

---

> ### Author Rebuttal · Authors · 2024-08-07
>
> Thank you for your thorough review and feedback. Below is a detailed point-by-point response addressing your main concerns and questions.
>
>
> > Explanation of why ST$_k$ outperforms AT$_k$.
>
> * ST$_k$ and AT$_k$ are not equivalent. Reference [1] proves the equivalence between AT$_k$ and MAT$_k$, as stated in Eq. (5).
> * The parameter $k$ is a hyper-parameter, usually we set $k$ to 0.9 $\times$ batch size, which is a practical choice. However it's impossible to determine the optimal $k$ for the model.
> * During the optimization process, once $k$ is set, AT$_k$ strictly (and harshly) filters out all samples ranked beyond the ($k$+1)-th position. In contrast, ST$_k$, which uses SGD for optimization, does not always have $\lambda$ as the k-th largest individual loss. Therefore, in the optimization process, $\lambda$ also serves as a parameter aimed at minimizing the overall loss.
>
> [1] Ogryczak, Wlodzimierz, and Arie Tamir. "Minimizing the sum of the k largest functions in linear time." Information Processing Letters 85.3 (2003): 117-122.
>
>
> > Lack of experiments on large-scale data, like ImageNet.
>
> ImageNet is a highly balanced dataset, with nearly the same number of images in each of its 1,000 classes. It is not a long-tailed classification dataset. We conduct experiments on its long-tailed version, ImageNet-LT.
>
> In addition to this, we also conduct experiments on 2 machine translation datasets. Due to the disparity in word frequencies, machine translation datasets are naturally long-tailed.
>
>
> > Is the ST$_k$ loss complementary to current state-of-the-art long-tailed learning methods.
>
> ST$_k$ represents an improvement in the aggregation method of individual losses, making it complementary to almost all existing long-tailed learning methods.
>
> PaCo/GPaCo/BCL, after obtaining a pre-trained model through contrastive learning, can utilize ST$_k$ to aggregate cross-entropy individual losses during subsequent transfer learning.
>
>
> > Ablation for the optimization manner of model parameters and $\lambda$
>
> In this work, we use Adam optimizer. We will consider including SGD or BCD (Block Coordinate Descent) as ablation studies.

---

> > ### Comment · Reviewer_46Ey · 2024-08-12
> > **Thanks for the responses from the authors**
> >
> > Thanks for the responses from the authors. I still have the following confusion on the paper.
> >
> > Q1. As the authors claim that $ST\_{k}$ can dynamically adjust the k for loss calculation, there should be some empirical or theoretical analysis to confirm the hypothesis.
> >
> > Q2. The proposed loss function is general and should be applicable to balanced data, like ImageNet.
> >        If not, the authors should provide more analysis on why the loss is specific to imbalanced data. How does the proposed method alleviate the overfitting issue of imbalanced data?
> >
> > Q3. The authors claim that the $ST\_{k}$ complements other long-tailed algorithms. However, there is no empirical analysis to support it.
> >
> > Q4. Why is the $ST\_{k}$ specific to the Adam optimizer?
> >
> > If the concerns can't be addressed, I prefer to keep the initial rating.

---

> ### Author Response · Authors · 2024-08-11
>
> Thanks again for reviewing our work. As the author-reviewer discussion phase wraps up, could you let us know if our responses have addressed your concerns? If so, would you reconsider your rating? If you still have concerns, please share so we can address them.

---

> ### Author Response · Authors · 2024-08-12
>
> Thanks for listing out all your confusions! Some of them can be explained theoretically, while others can be clarified through additional experiments. Here is the point-by-point response.
>
> > Q1. As the authors claim that ST$_k$ can dynamically adjust the k for loss calculation, there should be some empirical or theoretical analysis to confirm the hypothesis.
>
> * The $\lambda^* = \ell_{[k]}$ not only serves as a filter, excluding individual losses that smaller than $\ell_{[k]}$ (see $\min_\lambda k\lambda + \sum_{i=1}^n [\ell_i - \lambda]_+$). The $\lambda$ itself, also serves as a trainable parameter aiming to minimize the loss.
>
> * By smoothing the $[\cdot]_+$ function, the optimization process becomes stable and efficient (see standard division in Table 2, 3 and 5, also time cost in Table 1).
>
>
> > Q2. The proposed loss function is general and should be applicable to balanced data, like ImageNet. If not, the authors should provide more analysis on why the loss is specific to imbalanced data. How does the proposed method alleviate the overfitting issue of imbalanced data?
>
> * In this paper, we use the ST$_k$ module to smooth the Average Top-k Loss (AT$_k$ Loss) to demonstrate that ST$_k$ can be applied in deep learning scenarios. We did not conduct experiments on balanced datasets because: AT$_k$ Loss was designed for imbalanced datasets [1], as this was proved by its creator, empirically and theoretically.
>
> * The application of ST$_k$ does not require additional computational time/resources while consistently bringing performance improvements.
>
> [1] Lyu S, Fan Y, Ying Y, et al. Average top-k aggregate loss for supervised learning[J]. IEEE transactions on pattern analysis and machine intelligence, 2020, 44(1): 76-86.
>
>
> > Q3. The authors claim that the ST$_k$ complements other long-tailed algorithms. However, there is no empirical analysis to support it.
>
> This is a very valuable suggestion, glad that you asked! The fact is that, we did conduct experiments on “how ST$_k$ complements other long-tailed algorithms.” However, due to the limited space, we did not include them in the present version. Here we would like to  provide a temporary experiment, and consider providing a detailed ablation study in our final version.
>
> The backbone here is an Vision Transformer (ViT) pre-trained by MAE / CLIP [2][3]. "CS" here represents the cost-sensitive learning; while PEL was provided by [4], which is SOTA method on 2 long-tailed classification leaderboards. If the model was pre-trained by MAE, we add an Linear Classifier after the backbone; other-wise we follow the default PEFT in [4]. The batch size was set to 2048, training 30,000 steps. The results are as follows:
>
> Pretrained By| CS | PEL | ST$_k$ | ImageNet-LT | CIFAR-100-LT |
> |-------|-------|-------|-----------|-------------------|---------------------|
> | MAE| –     | –      | –          | 65.701           | 78.572             |
> | MAE| ✓     | –      | –         | 69.884           | 82.135             |
> | MAE| ✓     | –      | ✓         | 70.140           | 83.048             |
> | CLIP| –      | ✓     | –          | 78.296           | 89.103             |
> | CLIP| –      | ✓     | ✓         | 79.148           | 89.833             |
>
> [2] He K, Chen X, Xie S, et al. Masked autoencoders are scalable vision learners[C]//Proceedings of the IEEE/CVF conference on computer vision and pattern recognition. 2022: 16000-16009.
>
> [3] Radford A, Kim J W, Hallacy C, et al. Learning transferable visual models from natural language supervision[C]//International conference on machine learning. PMLR, 2021: 8748-8763.
>
> [4] Shi J X, Wei T, Zhou Z, et al. Parameter-efficient long-tailed recognition[J].
>
>
> > Q4. Why is the ST$_k$ specific to the Adam optimizer?
>
> * We claimed that the proposed loss can be optimized by the most common optimizers like SGD / Adam, only to stress the fact that no  additional time is needed to update $\lambda$ and the model parameters $w_{model}$ separately in two steps by using BCD (Block Coordinate Descent).
>
> * Empirically, ST$_k$ is not specific to Adam, it also works with SGD, AdaGrad, and BCD. The Supplementary Material we uploaded include the source code for the experiments. If necessary, we would consider providing the results of different optimizers in our final version.
>
>
> Hope these explanations address your concerns. If you have any confusion, please feel free to reach out to us anytime.

---

> ### Author Response · Authors · 2024-08-13
>
> We carefully reconsidered your questions. We would like to further clarify our intentions.
>
> **For your questions:**
>
> > Q2. The proposed loss function is general and should be applicable to balanced data, like ImageNet. If not, the authors should provide more analysis on why the loss is specific to imbalanced data. How does the proposed method alleviate the overfitting issue of imbalanced data?
>
> * The curves in Figure 3 clearly reflect that as the degree of imbalance increases, the Average Loss continuously misguides the model to shift the decision boundary toward the minority class.
> * In contrast, AT$_k$ Loss prevent the model from shifting by ignoring those individual losses with smaller values.
> * However,  the discontinuity affected the stability of the optimization process of $\lambda$. Therefore, we need to find a
>  objective function,
>     * with continuous gradient;
>     * at the same time sufficiently approximate the Average Top-k Loss $\frac{k \lambda}{n} + \frac{1}{n}\sum_{i=1}^n [\ell_i - \lambda]_+$.
> * $ST_k$ happens to achieve this. Theoretically, the approximation error can be bounded by $\delta/2, \forall \lambda, w_{model}$. This is a **uniform convergence**, which is known as a very strong condition. (because $\delta \in \mathbb{R}$, we call it point-wise convergence, this may cause some misunderstanding)
> * With the help of $ST_k$, models show consistent performance improvements, on synthetic datasets and real-world datasets.

---

### Official Review · Reviewer_uCvC · 2024-07-12

**Soundness:** 2
**Presentation:** 2
**Contribution:** 2
**Rating:** 4
**Confidence:** 1

**Summary:**

The authors proposed a differentiable layer to approximate the top-k loss in deep learning. The proposed layer is motivated from Eq. (1), replacing the ReLU with a smoothed ReLU (SReLU in Eq. (2)). The authors showed that the proposed layer is point-wise convergent to top-k loss. Numerical experiments validate the superiority of the proposed method.

**Strengths:**

* The paper is over-all well written, easy to follow. Experiments on real-world datasets are provided, in addition to simulated results.
* A fast, accurate approximate of top-k loss is important in deep ranking problems.
* Experiments show that the proposed method is over-all better than baseline methods.

**Weaknesses:**

* The biggest concern is that most of the real-world experiments are done on small scale datasets. It would be more convincing if the authors could demonstrate their method on large-scale ranking problems, especially, large deep learning models.

* The proposed method lacks strong theoretical guarantees. Point-wise convergence is somehow weak. As easy to expect, the convergence rate and error bound of the proposed method should depend on the data distribution, which is not discussed in the paper. It would be nice if the authors could show that under what conditions, the proposed method will converge faster than $AT_k$ and / or with smaller approximation error.

* It is important to check how well the proposed method approximates top-k loss in real-world problems. However, most real-world experiments report model accuracy, which is an indirect metric of top-k loss approximation. Please consider to measure the top-k loss approximation error directly.

**Questions:**

* In real-world problems, is it possible to design experiments to measure the top-k loss approximation error directly?
* What is the theoretical advantage of $ST_k$ v.s. $AT_k$

---

> ### Author Rebuttal · Authors · 2024-08-07
>
> We appreciate your feedback.
>
>
> **For your concerns:**
>
> > Limited dataset scale.
>
> We hold a different view on this.
>
> * We conduct experiments on datasets such as ImageNet-LT and Places-LT, which, to the best of our knowledge, are the largest unbalanced visual classification datasets available.
> * Additionally, we conduct experiments on two machine translation tasks using the Transformer-Base model, which has 0.11B trainable parameters, approximates that of Large Language Models (LLMs).
> * In recent years, as model sizes have become increasingly larger, some works that contain large-scale experiments have become hard to follow for the community. We advocate for conclusions that can be validated at appropriate sizes without the need for validation at the level of LLMs.
>
>
> > Point-wise convergence is theoretically weak, expect more convincing experimental evidence.
>
> * In Table 1, ST$_k$ not only guides the model to achieve higher accuracy but also converges faster compared to AT$_k$.
> * we experiment with the time cost of a few common sorting algorithms, AT$_k$ and ST$k$ to calculate the ranking average. The specific scheme of the experiment is to find the Top-k (k=5) sum from 10,000 normally distributed samples. For AT$_k$ and ST$_k$, we iterate until the error is less than $10^{-2}$, and for each algorithm, we conduct 50 experiments and record the average time taken in the third column of the Table.
>
> | Algorithm    | Complexity         | Average Time(s) |
> |--------------|--------------------|-----------------|
> | BubbleSort | $\mathcal{O}(n^2)$ | 20.42196$\pm$3.7015        |
> | HeapSort    | $\mathcal{O}(n\log(n))$ | 0.1243$\pm$0.0446   |
> | AT$_k$        | $\mathcal{O}(n+k)$ | 0.2167$\pm$0.1528     |
> | ST$_k$(Ours)| $\mathcal{O}(n+k)$ | **0.0127**$\pm$0.0020     |
>
> In experiments, ST$_k$ is the fastest algorithm to solve the ranking average value, on the other hand, AT$_k$ lacks robustness in training due to the gradient discontinuity.
>
>
> > Consider measuring the top-k loss approximation error directly.
>
> Unlike synthetic datasets, real-world problems do not have an explicit theoretical decision boundary. Therefore, we cannot provide a measure like the ParaF1 score in Table 1, which assesses the overlap between theoretical decision boundary and its estimation.
>
> We will consider including Precision, Recall, or F1-score in the final version, but these metrics would only serve as a supplement to Accuracy which cannot directly show the approximation performance.
>
>
> **For your Questions:**
>
>
> > In real-world problems, is it possible to design experiments to measure the top-k loss approximation error directly?
>
> Here comes a more fundamental question: in real-world classification problems, like ImageNet-classification, does an optimal model exist that minimizes the loss?
>
> Even if such an optimal model exists, it's likely that we cannot compute it with our finite resources, which is precisely why algorithms like Gradient Descent are so valuable. They allow us to approximate the optimal solution efficiently without the need to exhaustively search all possibilities.
>
>
> > What is the theoretical advantage of ST$_k$  v.s. AT$_k$
> ST$_k$ converges faster theoretically.

---

> ### Author Response · Authors · 2024-08-12
>
> Thanks for your insightful suggestions. Hope our explanations address your concerns. Based on the feedback, we are glad to inform you that we include 2 additional experiments.
>
> **1.** We present a temporary ablation over the values of the smooth coefficient $\delta$ on CIFAR-100-LT. All other settings remain the same as those in the paper, with only $\delta$ being changed.
>
> | $\delta$   | 0.1 | 0.05 | 0.01 | 0.005 |
> |--------------|-------|--------|-------|--------|
> | Accuracy | 88.8 | 89.4 | 89.8 | 89.8 |
>
> **2.** We conduct experiments on “how ST$_k$ complements other long-tailed algorithms.”
> The backbone here is an Vision Transformer (ViT) pre-trained by MAE / CLIP [1][2]. "CS" here represents the cost-sensitive learning; while PEL was provided by [4], which is SOTA method on 2 long-tailed classification leaderboards. If the model was pre-trained by MAE, we add an Linear Classifier after the backbone; other-wise we follow the default PEFT in [3]. The batch size was set to 2048, training 30,000 steps. The results are as follows:
>
> Pretrained By| CS | PEL | ST$_k$ | ImageNet-LT | CIFAR-100-LT |
> |-------|-------|-------|-----------|-------------------|---------------------|
> | MAE| –     | –      | –          | 65.701           | 78.572             |
> | MAE| ✓     | –      | –         | 69.884           | 82.135             |
> | MAE| ✓     | –      | ✓         | 70.140           | 83.048             |
> | CLIP| –      | ✓     | –          | 78.296           | 89.103             |
> | CLIP| –      | ✓     | ✓         | 79.148           | 89.833             |
>
> **[1]** He K, Chen X, Xie S, et al. Masked autoencoders are scalable vision learners[C]//Proceedings of the IEEE/CVF conference on computer vision and pattern recognition. 2022: 16000-16009.
>
> **[2]** Radford A, Kim J W, Hallacy C, et al. Learning transferable visual models from natural language supervision[C]//International conference on machine learning. PMLR, 2021: 8748-8763.
>
> **[3]** Shi J X, Wei T, Zhou Z, et al. Parameter-efficient long-tailed recognition[J].
>
>
> **3.** The explanations we provided to Reviewers `SwHE` and `46Ey` (we appreciate the concerns they raised) may appropriately answer your question **"What is the theoretical advantage of ST$_k$ vs AT$_k$?"**
>
> * The motivation here is that the discontinuity has affected the stability of the parameter $\lambda$ optimization process.
> Therefore, we need to find a smooth objective function to approximate: $\min_\lambda \frac{k \lambda}{n} + \frac{1}{n}\sum_{i=1}^n [\ell_i - \lambda]_+$
>
> * Other ReLU-variants cannot sufficiently approximate ReLU as they are only similar in shape. Therefore, we design SReLU and proved that the approximation error can be bounded by $\delta / 2$ (see page 12, A.1).
>
>   $ \left[ \frac{k \lambda}{n} + \frac{1}{n}\sum_{i=1}^n [\ell_i - \lambda]_+ \right] $
>
>   $- \left[ \frac{k \lambda}{n} + \frac{1}{n}\sum_{i=1}^n \frac{1}{2}\left[(\ell_i-\lambda) + \delta \left( \sqrt{\frac{(\ell_i-\lambda)^2}{\delta^2}+1}-1 \right)\right] \right]$
>
>   $= \sum_{i=1}^n \frac{1}{2n}  \left[ \sqrt{(\ell_i-\lambda)^2} - \sqrt{(\ell_i-\lambda)^2 + \delta^2} + \delta \right]$
>
>   $= \sum_{i=1}^n \frac{1}{2n}  \left[ \delta - \frac{\delta^2}{\sqrt{(\ell_i-\lambda)^2} + \sqrt{(\ell_i-\lambda)^2 + \delta^2}} \right]$
>
>   $=  \sum_{i=1}^n \frac{1}{2n}  \left[ \delta \left( 1 - \frac{1}{\sqrt{\frac{(\ell_i-\lambda)^2}{\delta^2}} + \sqrt{\frac{(\ell_i-\lambda)^2}{\delta^2} + 1}} \right) \right] < \frac{\delta}{2}$
>
> * The smoothed objective function shows jointly convex w.r.t $(w_{model}, \lambda)$, making ST$_k$ a special case of the non-linear multiple choice knapsack problems, with at most $q=2$ roots, which can be found in constant time. Thus, the problem can be solved in $\mathcal{O}(n*lnq)=\mathcal{O}(n)$ time as $q$ being fixed [4] (section 4, "non-linear case").
>
> * Experiments have shown that, empirically, we don't even need to spend additional time optimizing $\lambda$ separately. Instead, by optimizing it synchronously with the model parameters $w_{model}$, we can still achieve performance improvements.
>
>    **[4]** Zemel E. An O (n) algorithm for the linear multiple choice knapsack problem and related problems[J]. Information processing letters, 1984, 18(3): 123-128.
>
>     **[5]** N. Megiddo, Linear programming in linear time when the dimension is fixed, J. ACM 31 (1984) 114–127.
>
>     **[6]** Ogryczak, Wlodzimierz, and Arie Tamir. "Minimizing the sum of the k largest functions in linear time." Information Processing Letters 85.3 (2003): 117-122.
>
> * By smoothing the $[\cdot]_+$ function, the optimization process becomes stable and efficient (see standard division in Table 2, 3 and 5, also time cost in Table 1).

---

> ### Author Response · Authors · 2024-08-13
>
> Thanks again for reviewing our work. After carefully considering all your suggestions, we realized there is one more thing we need to clarify.
>
> We hold a different view on your concern that **"point-wise convergence is somehow weak."** The approximation error between the smoothed loss function and the original loss function can uniformly bounded by $\delta/2, \forall (\lambda, w_{model})$ (see page 12, A.1), which means this is the **uniform convergence**, a very strong condition.
>
> We sincerely hope you may reconsider whether our response fully addresses your concern.
>
> Best wishes,
> Authors.

---

### Official Review · Reviewer_8VJU · 2024-07-12

**Soundness:** 3
**Presentation:** 3
**Contribution:** 3
**Rating:** 7
**Confidence:** 3

**Summary:**

Authors introduce a novel differentiable module (ST_k) for efficiently solving top-k problems. Their method relies on optimizing a differentiable form of an equivalent optimization problem, by proposing an approximation of ReLU differentiable everywhere. This equivalent optimization problem contains a single parameter \lambda which is supposed to match the k-th largest element in the optimal case, and in practice is optimized over the dataset. Authors  experiment on synthetic data, showing that their loss objective results in a model that more closely matches the true decision boundary in this setting. Next, authors experiment with real-world applications on binary classification and long-tailed classification datasets, showing consistent improvement over other optimization losses.

**Strengths:**

**Strengths**

- Authors propose an efficient differentiable solution to the Top-K problem that integrates very easily into existing model architectures, and only has a single optimizable parameter.
- Authors show consistent improvements both in synthetic and real-world tasks compared to sota methods.
- The paper is well-written, authors provide ample motivation for their design choices.

**Weaknesses:**

**Weaknesses**

- Although the improvements shown by the authors are consistent, they are also marginal - only improving slightly in for example CIFAR-100-LT classification and ImageNet-LT classification compared to average aggregation.
- As authors mention, they only apply their module to a single Top-K Loss; ATK, making it unclear how ST_k could be applied in other scenarios and how well it generalizes to different settings.

**Questions:**

**Questions**
- How do you tune the smoothing coefficient in practice? Is this a hyperparameter that needs to be sweeped over? That would somewhat deteriorate the efficiency of your module. I would like to see an ablation over the values of delta, i think it is important to see what/if any impact this choice has on model performance.
- Did you experience any training instabilities with the two-step optimization scheme?

**Limitations:**

The authors mention as limitation the fact that they only applied ST_k to the average top-k loss. I think it would be good if they more explicitly mention future research directions that they are interested in/ that would be of value (there is space in the manuscript).

---

> ### Author Rebuttal · Authors · 2024-08-07
>
> Your positive feedback is very encouraging.
>
> **For your concerns:**
>
> > Although the improvements shown by the authors are consistent, they are also marginal - only improving slightly in for example CIFAR-100-LT classification and ImageNet-LT classification compared to average aggregation.
>
> We acknowledge that the performance improvement brought by the smoothed Average Top-k Loss is not significant.
>
> * In the field of Computer Vision, classification accuracy is an extensively explored direction. In fact, any improvement is a further advancement beyond a high point.
> * By incorporating a single trainable parameter into the computation graph, ST$_k$ achieves performance enhancements without requiring additional computational resources or time, thereby making the proposed module energy-efficient.
> * Additionally, the adjustment to the aggregation method of the individual losses is compatible with almost all existing methods for handling imbalanced data, which is likely to attract widespread attention from the community focused on improving classification accuracy.
>
>
> > As authors mention, they only apply their module to a single Top-K Loss; ATK, making it unclear how ST_k could be applied in other scenarios and how well it generalizes to different settings.
>
> ST$_k$ can be applied end-to-end to models in any scenario that involves Top-K problem (see page 3, lines 87-89).
>
>
> **For your questions:**
>
> > How do you tune the smoothing coefficient $\delta$ in practice? Is this a hyperparameter that needs to be sweeped over? That would somewhat deteriorate the efficiency of your module. I would like to see an ablation over the values of delta, i think it is important to see what/if any impact this choice has on model performance.
>
> * The smoothing coefficient $\delta$, 0.01 is a grid search determined value for all datasets in section 5.1 (see page 7, line 186) and was adopted in all the other experiments.
> * We believe that the ablation study is necessary and will include it in our final version. Thanks for the insightful suggestion.
>
>
> > Did you experience any training instabilities with the two-step optimization scheme?
>
> * As mentioned above (see page1, line 25), in the experiments conducted in this paper, ST$_k$ does not require a two-step optimization using block coordinate descent, but can synchronously update $\lambda$ and model parameters using SGD/Adam.
> * Due to the elimination of points where gradients vanish, the training process is very stable, as evidenced by the standard deviations listed in Tables 2, 3, and 5.

---

> > ### Comment · Reviewer_8VJU · 2024-08-09
> >
> > I thank the authors for their response. My concerns have been answered, i i crease my recommendation.

---

> ### Author Response · Authors · 2024-08-10
>
> We really appreciate your recommendation!
>
> Here we present a temporary ablation over the values of the smooth coefficient $\delta$ on CIFAR-100-LT. All other settings remain the same as those in the paper, with only $\delta$ being changed.
>
> | $\delta$   | 0.1 | 0.05 | 0.01 | 0.005 |
> |--------------|-------|--------|-------|--------|
> | Accuracy | 88.8 | 89.4 | 89.8 | 89.8 |
>
> Integral experiments will be included in our final version. Thanks again for your suggestions.

---

### Author Rebuttal · Authors · 2024-08-07

We appreciate all your comments.

Here is a summary of the strengths and a general response to the concerns received:

**Strengths:**

Firstly, we appreciate that all reviewers think our paper to be clearly written and easy to follow.
- Reviewer 8VJU noted that our framework is efficient and integrates very easily into existing model architectures, and believed that our motivation was ample.
- Reviewer uCvC found our method to be fast and overall better than baseline methods.
- Reviewer 46Ey thought our method was flexible and is expected to be used in many scenarios.
- Reviewer SwHE observed that we conducted extensive experiments, using both synthetic and real-world datasets.

**Concerns:**

> The limited scale of the dataset:

* We conduct experiments on datasets such as ImageNet-LT and Places-LT, which, to our knowledge, are among the largest imbalanced visual classification datasets available.
* Additionally, we have conducted experiments on two machine translation tasks using the Transformer-Base model, which has 0.11B trainable parameters.
* In recent years, as model sizes have become increasingly larger, some works that contain large-scale experiments have become hard to follow for the community. We advocate for conclusions that can be validated at appropriate sizes without the need for validation at the level of LLMs.

> To add more ablation studies and time-cost experiments:

We add 2 additional experiments during rebuttal and will consider including more experiments in our final version.

---

### Author Response · Authors · 2024-08-14
**General Response after Author-Reviewer Discussion**

We are very grateful for the efforts and suggestions of each reviewer during the author-reviewer discussion period. This has greatly helped in improving the quality of this work. Below are the supplements and improvements we made based on the Reviewer feedback.

**1. We added 3 experiments temporarily.**

* We present a temporary ablation over the values of the smooth coefficient $\delta$ on CIFAR-100-LT. All other settings remain the same as those in the paper, with only $\delta$ being changed.

| $\delta$   | 0.1 | 0.05 | 0.01 | 0.005 |
|--------------|-------|--------|-------|--------|
| Accuracy | 88.8 | 89.4 | 89.8 | 89.8 |

* We conduct experiments on “how ST$_k$ complements other long-tailed algorithms.”
The backbone here is an Vision Transformer (ViT) pre-trained by MAE / CLIP [1][2]. "CS" here represents the cost-sensitive learning; while PEL was provided by [4], which is SOTA method on 2 long-tailed classification leaderboards. If the model was pre-trained by MAE, we add an Linear Classifier after the backbone; other-wise we follow the default PEFT in [3]. The batch size was set to 2048, training 30,000 steps. The results are as follows:

Pretrained By| CS | PEL | ST$_k$ | ImageNet-LT | CIFAR-100-LT |
|-------|-------|-------|-----------|-------------------|---------------------|
| MAE| –     | –      | –          | 65.701           | 78.572             |
| MAE| ✓     | –      | –         | 69.884           | 82.135             |
| MAE| ✓     | –      | ✓         | 70.140           | 83.048             |
| CLIP| –      | ✓     | –          | 78.296           | 89.103             |
| CLIP| –      | ✓     | ✓         | 79.148           | 89.833             |

  **[1]** He K, Chen X, Xie S, et al. Masked autoencoders are scalable vision learners[C]//Proceedings of the IEEE/CVF conference on computer vision and pattern recognition. 2022: 16000-16009.

  **[2]** Radford A, Kim J W, Hallacy C, et al. Learning transferable visual models from natural language supervision[C]//International conference on machine learning. PMLR, 2021: 8748-8763.

  **[3]** Shi J X, Wei T, Zhou Z, et al. Parameter-efficient long-tailed recognition[J].

* We experiment with the time cost of a few common sorting algorithms, $AT_k$ and $ST_k$ to calculate the ranking average. The specific scheme of the experiment is to find the Top-k (k=5) sum from 10,000 standard normally distributed samples. For $AT_k$ and $ST_k$, we iterate until the error is less than $10^{-2}$, and for each algorithm, we conduct 50 experiments and record the average time taken in the third column of the Table.

| Algorithm | Complexity       | Average Time(s)          |
|-----------|------------------|--------------------------|
| BubbleSort | $\mathcal{O}(n^2)$     | 20.42196 ± 3.7015        |
| HeapSort   | $\mathcal{O}(n \log(n))$ | 0.1243 ± 0.0446          |
| $AT_k$   | $\mathcal{O}(n + k)$     | 0.2167 ± 0.1528          |
| $ST_k$ (Ours) | $\mathcal{O}(n + k)$ | **0.0127 ± 0.0020**      |


**2. We realize that our current wording may causes some confusion.**

The approximation error between the smoothed loss function and the original loss function can uniformly bounded by $\delta/2, \forall (\lambda, w_{model})$ (see page 12, A.1), which means this is the uniform convergence, a recognized very strong condition.



**Thanks again for reviewing our work. We are grateful for the valuable feedback provided by each reviewer. It is a great honor to meet you at NeurIPS2024.**

---

### Decision · Program_Chairs · 2024-09-25

**Decision:**

Accept (poster)

**Comment:**

The paper received the following ratings:  R1 - 7 (accept), R2 - 4 (borderline reject, no reaction to rebuttal), R3 - 4 (borderline reject), R4 - 6 (weak accept).

Overall reviewers highlighted that paper is clearly written and easy to follow, that the proposed top-k loss is an important problem, and that the experiments are clear and show the improvement by the provided loss.

The main weaknesses listed are that the evaluation might be a bit limited and that more theoretical would be helpful. The authors provided a solid rebuttal which provided additional information.
While I share the concerns of the reviewers for the original paper, I think that the paper can significantly improve if all comments and additional experiments from the rebuttal would be incorporated in the main paper and/or the supplementary. Given the rebuttal, I'm also confident that the authors are interested in improving the paper and will do this.

I therefore recommend accepting the paper.